# Central and Peripheral Immune Dysregulation in Posttraumatic Stress Disorder: Convergent Multi-Omics Evidence

**DOI:** 10.3390/biomedicines10051107

**Published:** 2022-05-10

**Authors:** Diana L. Núñez-Rios, José J. Martínez-Magaña, Sheila T. Nagamatsu, Diego E. Andrade-Brito, Diego A. Forero, Carlos A. Orozco-Castaño, Janitza L. Montalvo-Ortiz

**Affiliations:** 1Department of Psychiatry, Yale University School of Medicine, New Haven, CT 06510, USA; diana.nunez@yale.edu (D.L.N.-R.); jose.martinez-magana@yale.edu (J.J.M.-M.); sheila.nagamatsu@yale.edu (S.T.N.); diego.andrade@yale.edu (D.E.A.-B.); 2VA CT Healthcare Center, West Haven, CT 06516, USA; 3Health and Sport Sciences Research Group, School of Health and Sport Sciences, Fundación Universitaria del Área Andina, Bogotá 110231, Colombia; dforero41@areandina.edu.co (D.A.F.); corozco35@areandina.edu.co (C.A.O.-C.)

**Keywords:** PTSD, immune system, epigenetic, transcriptomic, multi-omic, brain, peripheral tissues, human, animal models

## Abstract

Posttraumatic stress disorder (PTSD) is a chronic and multifactorial disorder with a prevalence ranging between 6–10% in the general population and ~35% in individuals with high lifetime trauma exposure. Growing evidence indicates that the immune system may contribute to the etiology of PTSD, suggesting the inflammatory dysregulation as a hallmark feature of PTSD. However, the potential interplay between the central and peripheral immune system, as well as the biological mechanisms underlying this dysregulation remain poorly understood. The activation of the HPA axis after trauma exposure and the subsequent activation of the inflammatory system mediated by glucocorticoids is the most common mechanism that orchestrates an exacerbated immunological response in PTSD. Recent high-throughput analyses in peripheral and brain tissue from both humans with and animal models of PTSD have found that changes in gene regulation via epigenetic alterations may participate in the impaired inflammatory signaling in PTSD. The goal of this review is to assess the role of the inflammatory system in PTSD across tissue and species, with a particular focus on the genomics, transcriptomics, epigenomics, and proteomics domains. We conducted an integrative multi-omics approach identifying TNF (Tumor Necrosis Factor) signaling, interleukins, chemokines, Toll-like receptors and glucocorticoids among the common dysregulated pathways in both central and peripheral immune systems in PTSD and propose potential novel drug targets for PTSD treatment.

## 1. Introduction

Posttraumatic stress disorder (PTSD) is a chronic and multifactorial disorder with a prevalence ranging between 6–10% in the general population, and 35% among individuals with high-lifetime trauma exposure (e.g., combat veterans) [1,2,3]. PTSD can develop after experiencing traumatic events; however, not all individuals exposed to trauma are diagnosed with PTSD, and this is known as resilience [4]. That differential response to trauma unmasks the existence of biological processes contributing to PTSD risk [1,2,3,5]. Growing evidence indicates that the immune system may contribute to the development, maintenance, and clinical outcomes of PTSD, establishing the dysregulation of the immune system as a hallmark feature of PTSD [6,7]. By examining 65 published studies in PTSD, one study summarized a comprehensive list of dysregulated inflammatory factors reported in the blood and cerebrospinal fluid samples of PTSD patients [7]. The dysregulation of immune cells, the human leukocyte antigen (HLA), and immune-related genes in PTSD also support the important role of the immune system in PTSD [6]. While the immune dysregulation in PTSD is a well-established finding in the literature, the differences between the central and peripheral immune system, its crosstalk, and the biological mechanisms underlying this dysregulation remain poorly understood.

The role of the hypothalamic-pituitary-adrenal (HPA) axis on the immune response is one of the most described physiological mechanisms in PTSD, impacting and linking both the central and peripheral systems. Briefly, after exposure to a traumatic event, the HPA axis is activated and induces a release of glucocorticoid hormones (GCs) [8,9], which, in turn, can activate glucocorticoid receptors (GR) and subsequently regulate the gene expression of inflammatory molecules, such as cytokines and the nuclear factor kappa B (NF-κB) [8,10]. In the periphery, GCs can regulate the response of innate immune cells, including the monocytes, macrophages, and dendritic cells [10]. In the brain, abnormal levels of GCs can impact the regulatory feedback of the HPA axis (known as glucocorticoid resistance), followed by an increased neuroinflammatory response, glial dysfunction and synaptic loss [9,11,12,13,14,15]. An animal model study found that pro-inflammatory cytokines in central and peripheral tissues are released after activation of central fear circuits and suggests that neuroinflammation can promote peripheral inflammation, migration of peripheral monocytes to the brain and the subsequent increased activity of microglia [16]. Neuroinflammatory processes orchestrated by central and peripheral systems [17] are mediated by microglia, brain cells also related to disturbances in fear memory in animal models [18,19] and PTSD [20]. These studies support an interplay between central and peripheral immune systems in PTSD and highlight the utility of animal models to shed light on the immunological mechanisms implicated in PTSD [21,22,23,24].

In recent years, high-throughput methods have facilitated genomic, transcriptomic, epigenomic, and proteomic evaluation to identify the mechanisms of dysregulated pathways in both central and peripheral tissues from humans with PTSD, as well as animal models of PTSD [2,25,26]. The noticeable dysregulation of the immune system in PTSD has also been also supported at different regulatory levels. For instance, the C-reactive protein (CRP) that participates in the activation of the complement system is frequently elevated in individuals with PTSD and is related with symptom severity [27,28,29,30]. At the genomic level, genetic variants such as rs1130864, rs3091244, rs1205 and rs2794520 identified through genome wide association studies (GWAS) have been associated with variable levels of CRP in individuals with PTSD [6,27,30]. Another immunoregulatory molecule reported with multi-omic dysregulation is *FKBP5* (FKBP Prolyl Isomerase 5, which mediates the GR translocation to the nucleus) with the associated genetic risk variants [31], alterations in DNA methylation patterns [32] and dysregulated expression in different brain regions [2,33,34].

This systematic review summarizes the recent evidence of multi-omics studies to dissect the role of the central and peripheral immune dysregulation in PTSD. We highlight convergent inflammatory factors across tissues and species and examine whether gene regulatory mechanisms may orchestrate the system-wide immune dysregulation in PTSD. Furthermore, we use computational approaches to explore immunoregulatory targets as potential PTSD treatments.

## 2. Literature Search

The literature search in the PubMed electronic database was conducted in December 2021 to capture full-text studies that investigated the role of the immune system in central and peripheral tissues of individuals with or animal models of PTSD. To reduce the bias in the selection of studies, we followed the Preferred Reporting Items for Systematic Reviews analysis and Metal Analysis (PRISMA) criteria [35]. We found 77 full-text records using the following keywords and Boolean operators: (“Immune” OR “Inflammatory” OR “immunological”) AND (“PTSD” OR “Posttraumatic Stress Disorder” OR “Post-traumatic Stress Disorder”) AND (“Transcriptome” OR “Transcriptomic” OR “Epigenome” OR “Epigenomic” OR “Bisulfite” OR “Array” OR “RNA-Seq” OR “EWAS” OR “epigenome-wide association”). To carry out the records screening, our inclusion criteria were as follows: (1) full-text articles published in the last 10 years, (2) studies using genome-wide approaches and (3) studies using human samples or wild-type and drug-free animal models of PTSD. For this last criteria, wild-type and drug-free animal were selected as a criterion to reduce noise in the transcriptional/epigenetic results related to PTSD. Furthermore, among the animal stress paradigms, we selected studies utilizing physical stressors, such as immobilization, electric shock, predator stress, single prolonged stress and stress-enhanced fear learning. These stress paradigms have been shown to induce fear memory in animals and mimic PTSD-like behaviors (referred here as animal models of PTSD) [21,22,23,36]. Records screening is represented in Figure 1.

## 3. Results

### 3.1. Genetic Variants Associated with PTSD Targeting Immune-Related Genes

PTSD has an estimated heritability of ~49%, of which only 6–20% is explained by common variants [3,37,38]. GWAS studies have reported loci associated with an increased risk of developing PTSD, some of these mapping to immune-related genes [3,37,39,40]. In 2015, a GWAS study in 3494 males with European and African ancestries from military cohorts identified significant single nucleotide polymorphisms (SNPs) in two genes related to immune response, *JAK1* and *FASLG* [41]. In a targeted study focusing on the Human Leukocyte Antigen (HLA) genes in an African American cohort (*n* = 429), five SNPs were identified as mapping to *B*, *C*, *DRB1*, *DQA1*, *DQB1*, and the *DPB1* immune-related genes associated with PTSD. This study also examined gene expression modules of PTSD through weighted gene co-expression network analysis (WCGNA) and identified immune-related enriched pathways [42]. Another GWAS replicated the association of the *HLA-B* gene with PTSD, a gene essential in the antigen presentation to lymphocytes [3,43]. In the same year,. another group studied rare genetic variations in 707 adolescents exposed to a tornado who met PTSD criteria, and identified the immune system in the gene network pathway analysis [44]. By performing a variant calling analysis on an RNA-seq dataset in a small cohort of infantry soldiers *(n* = 85, 27 with PTSD symptoms), a study revealed a possible role of tumor necrosis factor (TNF) in PTSD based on the identified genes with high impact mutations observed in solders with PTSD symptoms [45]. Another study assessing (CRP) in 286 U.S. military veterans of post-9/11 conflicts reported an association between the rs3091244 SNP, lifetime trauma exposure, and PTSD severity [30].

Recent large-scale GWAS of PTSD from consortia efforts have also identified genome-wide significant SNPs located in immune-related genes. A study of 146,660 European Americans and 19,983 African Americans individuals from the Million Veteran Program (MVP) cohort assessing reexperiencing PTSD symptoms identified a significant association in the *RAB27B* gene, which plays a role in inflammatory response [39]. *PACRG* was later identified in 250,000 participants from the MVP of European and African ancestries [40]. *PARK2* was reported assessing the Psychiatric Genomics Consortium PTSD (PGC-PTSD) Freeze 2 dataset (PGC2) and replicating their results in the MVP cohort [3]. Both *PARCG* and *PARK2* genes are part of the parkin complex that participate in critical autophagic immune processes [3,40,46].

### 3.2. Transcriptomic and Epigenomic Dysregulation of Immune-Related Genes in PTSD

Studies have found that the dysregulation of immune-related genes, such as *IL-1A* and *FKBP5*, may be mediated by epigenetic mechanisms which, in turn, may result in up- or down-regulation of gene expression and the subsequent immune signaling dysfunction [2,27,29,31,33,34,47,48]. Here, we reviewed genome-wide transcriptomic and epigenomic studies of PTSD across tissues and species. We identified four studies in brain samples, one in human brains and three in animal models of PTSD [25,49,50,51,52]. From 22 studies in peripheral samples, 21 in human samples [6,45,50,51,53,54,55,56,57,58,59,60,61,62,63,64,65,66,67,68,69,70,71,72], and one in animal models of PTSD [50,51], only one study evaluated both peripheral and brain samples in a rodent model of PTSD and reported the immune dysregulated genes across tissues [50,51]. Twelve studies indicated the immune-related genes identified in their cohorts [25,49,50,51,52,54,58,59,61,68,71]. Seven studies published the whole list of dysregulated genes [45,55,57,60,62,69,73] from which immune-related genes were filtered out through GO enrichment analysis using STRING [74] and Metascape [75] resources. We retained all genes involved in immune signaling and conducted a comprehensive convergent analysis across peripheral and brain tissues in both human and animal models of PTSD.

#### 3.2.1. Peripheral Immune Dysregulation

Transcriptomic findings

Only one animal model of PTSD study has evaluated gene expression changes at the genome-wide level. This transcriptomic study examined the effects of aggressor exposure at different points (1, 10, and 42 days after exposure). The authors evaluated the hippocampus, amygdala, and medial prefrontal cortex (mPFC) in a 2015 study [50], and the hemibrain, blood and spleen samples in a follow-up 2017 study [51]. The identified DEGs across tissues and different time points evaluated were enriched in inflammatory pathways such as activation of leukocytes, adhesion of immune cells, cytokine activity, interleukins and (NF-κB) signaling [51]. These findings suggest a sustained neuroinflammatory response in PTSD [51]. 

In human samples, we identified 245 immune-related genes differentially expressed in PTSD. A research group conducted three transcriptomic studies on blood samples in 2017, 2019 and 2021. In the first study, they evaluated 324 World Trade Center responders and reported 448 differentially expressed genes (FDR < 0.05). Further, *PBRM1*, *PIK3CA*, *MED1*, *MAP3K1*, *JAK2*, *CEBPB*, *CD163*, *MED14*, *NCOA3*, *KAT2B*, *NFAT5*, *AKT1*, *PPP3CB*, *NCOR1* and *FKBP5* genes were nominally enriched in the glucocorticoid receptor signaling pathway [57]. In the second study, they conducted a transcriptomic analysis on unsorted blood cells as well as cell-type specific transcriptomic analysis in 39 World Trade Center responders (20 PTSD cases and 19 controls). They identified both cell-type specific and common DEGs across all cell types. In B cells, *ABCA6*, *ANEP*, *CHN2*, *IL-17RB*, *IL-7R*, *KIAA1217*, *TBC1D4*, *TESK2*, and *TGFBI* were differentially expressed in PTSD. *PIK3AP1 and SMPDL3A* were identified only in CD4T cells; *SMAD1* and *TLR5* in CD8T cells; and *HSF5*, *IPO5*, *REST*, *SLC38A1*, and *SLC44A2* in monocyte cells. *FKBP5* and *PI4KAP1* were upregulated across all cell types evaluated [68]. In a recent transcriptomic study assessing PTSD symptom-level data in 226 World Trade Center responders, they identified expression modules associated with inflammatory processes, such as neutrophil activation and the interferon signaling pathway [69].

A peripheral immune dysregulation in PTSD was also indicated by three transcriptomic studies from another research group. In the first study, dysregulation of the innate immune system and interferon signaling was identified in 188 U.S. Marines exposed to conflict zones [59]. On the same dataset, a deconvolution analysis performed to identify hub transcription factors orchestrating gene network functions (master regulators), indicated that *TNFAIP3*, *TRAFD1* and *PML* may act as master regulators affecting the immune signaling in PTSD [58]. Furthermore, a follow-up study integrated information of seven types of traumas from five independent PTSD blood transcriptome studies [60]. By using an integrative co-expression network analysis, they confirmed a dysregulation of the immune response and cytokine signaling across trauma types in PTSD [60].

The IL-6 is a cytokine with pro-inflammatory and anti-inflammatory pleiotropic effects in the immune system [76,77,78], and elevated serum levels of IL-6 have been reported in individuals with PTSD [79]. Transcriptomic changes related to varying serum levels of IL-6 were assessed by evaluating differential expression in Japanese civilian women with PTSD having high (*n* = 16) or normal (*n* = 16) IL-6 levels and health controls (*n* = 16) [54]. Women with PTSD and elevated levels of IL-6 showed differential expression of *IL-4*, *IL-18R1*, *IL-28RA*, *DEFA3*, *DEFA4*, *FCGR1A*, *FCGR1B*, *CEACAM8* and *LTF*, and an enrichment of the inflammatory signaling pathway. In contrast, women with PTSD and normal IL-6 levels showed DEGs enriched for neurotransmission and nervous system development [54]. These findings suggest a key role of immunological factor IL-6 in PTSD.

Differential expression of immune-related genes, particularly those involved in Th cell differentiation, was also indicated in blood samples of eight PTSD veterans compared with four healthy controls [61,72]. In a follow-up study, they showed that differential expression of *WNT10B* may be mediated by epigenetic changes (See the ‘Epigenomic’ section) [72]. Innate immune response, cytokine-cytokine receptor interaction, Jak-STAT and Toll-like receptor were additional enriched pathways reported in transcriptomic studies comparing veterans with and without PTSD [62,73]. Another study comparing military service members with high and low symptoms severity found a significant differential expression only in PTSD patients with high intrusion symptoms and an upregulation of the immune response related to the NF-κB hub. It suggested an implication of this pathway in intrusion symptoms of PTSD [55]. Taken together, findings from peripheral transcriptomic studies support a role of the immune dysregulation on PTSD and suggests cell-type specificity effects. Further, it provides evidence on how changes in gene expression may be mediated by epigenetic modifications.

Epigenomic findings

Epigenetics are chemical modifications at the DNA that influence the chromatin configuration and, subsequently, gene transcription processes. DNA methylation (DNAm) is one of the most common epigenetic modifications studied, defined as the addition of a methyl group in the 5′ position of a cytosine ring (5-methyl cytosine, 5mC). This mechanism is mainly associated with transcriptional repression when the 5mC occurs at the promoter region [2,55,79,80,81,82]. Studies in the last decade have evaluated epigenetic changes associated with PTSD at the genome-wide level through epigenome-wide association studies (EWAS). In our literature search, we identified six records examining peripheral samples of individuals with PTSD and evaluating blood and saliva samples.

One of the first EWAS of PTSD conducted in 100 trauma-exposed civilians (23 with PTSD) from the Detroit Neighborhood Health Study (DNHS) assessed methylation and non-methylation (unmethylated) and reported the unmethylation of *LTA4H*, *CXCL8*, *AQP9*, *TREM1*, *F8*, *CCL1*, *PYDC1*, *KLRG1*, *IFI35*, *CD1D*, *CD2*, *NLRP12*, *GBP1*, *IFI16*, *LST1*, *PTPN22*, *TLR1*, *TLR3*, *CMKLR1*, *STAP1* and *SLAMF7* in individuals with PTSD diagnosis, all immune-related genes. Another early EWAS study of PTSD (*n* = 25 cases and 25 controls) showed increased global DNA methylation in PTSD subjects with significant differential methylation in immune-related genes such as ANXA2 and TLR8 [64]. Epigenomic changes impacting immune signaling in PTSD were also found in two additional studies, but gene-level data was not provided [80,83]. Another meta-analysis from the PGC PTSD EWAS Workgroup (*n* = 545) identified cg19577098 covering the HGS gene, part of the endosome complex and related to interleukin 6 (IL-6) and Tumor Necrosis Factor alpha (TNF-α) signaling [69]. DNA methylation at cg10636246 targeting the Absent in Melanoma 2 (AIM2) gene was related to lower serum CRP levels and immune response in PTSD [30]. A large EWAS of PTSD performed by the PGC PTSD EWAS Workgroup in blood samples from ten military and civilian cohorts (*n* = 1896) reported four genome-wide significant CpG sites in the AHRR, a gene involved in the kynurenine metabolism and immune response [70]. In saliva, an EWAS of current and lifetime PTSD in 1135 male veterans from the National Health and Resilience Study cohort from our group identified the immune-related genes DYNC1H1 and AP2B1 (current PTSD), and CD55 (lifetime PTSD) [80,83].

EWAS studies of PTSD have also utilized a longitudinal study design evaluating pre- and post-deployment in male veteran cohorts, all examining blood tissue. By evaluating 429 subjects from three male military cohorts (US Marine Resiliency Study (MRS), US Army Study to Assess Risk (STARRS), and Resilience in Servicemembers (Army STARRS), a study reported an association of CpG in the immune-related gene F2R with both pre- and post-deployment PTSD symptoms severity [65]. A significant differentially methylated region (DMR) in HEXDC was associated with post-deployment in a longitudinal meta-EWAS of PTSD in 226 individuals from these three male veteran cohorts [72]. HEXDC is related to immune signaling in PTSD based on the association of this gene with rheumatoid arthritis [72,84]. Another significant PTSD DMR targeted the human leukocyte antigen complex proteins HLA-DPB1, HLA-DBP1, and HLA-DRB1 genes which are involved in immune processes and play a role in synaptic plasticity, learning memory and stress reactivity [72].

In addition to DNA methylation, histones modifications (e.g., methylation or acetylation) and micro-RNAs (inducing the degradation of mRNA) can also impact gene expression and have been associated with psychiatric disorders [85,86,87], including PTSD [85,88]. The role of these epigenetic mechanisms in the immune signaling and PTSD was demonstrated by Bam 2020 in blood samples of eight PTSD veterans [71]. They found that the expression of the immune-related gene WNT10B may be mediated by increased H3K4me3 (tri-methylation at the 4th lysine residue) at the promoter region and a downregulation of the miRNA hsa-miR-7113-5p [71].

An integrative multi-omic analysis can help disentangle the relationship between epigenetic modifications and gene expression and identify markers and mechanisms involved in complex psychiatric disorders [89,90,91,92,93]. To assess the convergent transcriptomic and epigenomic changes that may mediate the peripheral immune dysregulation in PTSD, we conducted an integrative multi-omic analysis across species. For transcriptomic studies, we found 69 immune-relate DEGs identified in more than one independent study (Appendix A). Transcriptional regulators, such as STAT (Signal Transducer and Activator of Transcription), IRF (interferon regulatory factors), Toll-like receptors (TLRs), and C1q proteins are well-known protein families involved in the innate and adaptive immune system [94,95,96,97]. Here, we identified the gene families from these immune proteins among the 69 convergent DEGs across species: Interleukins (IL-10, IL-16, IL-1RN, IL-2, and IL-6), interferon regulatory factors (IRF3 and IRF9), STATs proteins (STAT1, STAT2 and STAT4) and C1q proteins (C1QA and C1QB). When evaluating the overlap between the epigenomic and transcriptomic findings, we identified eight convergent genes: AQP9, WNT10B, GBP1, IFI35, NLRP12, CXCL8, TLR1 and TLR3. Our multi-omic exploration integrating epigenomic and transcriptomic findings in peripheral tissues helped identify convergent mechanisms for the immune dysregulation in PTSD and reveal potential therapeutic targets. In the next section, we review transcriptomic and epigenomic changes in brain samples of PTSD.

#### 3.2.2. Central Immune Dysregulation

Transcriptomic findings

A total of 415 unique DEGs were identified across the four brain transcriptomic studies examined in this review: three DEGs in human samples and 412 DEGs in animal models.

In animal models of PTSD, mitochondrial expression analysis in the amygdala identified 17 dysregulated genes related to inflammatory signaling [52]. Furthermore, another study evaluating transcriptional dysregulation across central and peripheral tissues in an animal model of PTSD reported DEGs enriched for inflammatory pathways, such as the activation of leukocytes, adhesion of immune cells, cytokine activity, interleukins and NF-κB signaling [50,51]. In our convergent analysis, we identified 12 convergent DEGs across the studies using rodent models of PTSD: *ADORA2A*, *ADORA3*, *ADRBK1*, *AKT1*, *ALS2*, *ANGPT2*, *ANXA2*, *AP3B1*, *APOE*, *APP*, *ARRB2* [49,50,51] and *GNB4* [50,51,52], which were related with immune signaling in PTSD. In humans, a transcriptomic study of advanced epigenetic age in PTSD examined the motor cortex, ventromedial, and dorsolateral prefrontal cortex (dlPFC) of 97 postmortem brains from the VA National PTSD Brain Bank [80]. Epigenetic age is a biological age indicator based on DNAm predictors of age-related morbidity and mortality of higher accuracy than chronological age [81]. This study identified *BMPER* (related to glucocorticoid pathway), *ALB* (oxidative stress), and *CCL19* (immune response) immune-related DEGs. Using a targeted approach, reduced expression of *IL-1A* was observed in the prefrontal cortex (PFC) of individuals with PTSD [82].

When evaluating the convergence between human and animal models of PTSD studies, we identified two immune-related DEGs: *IL-1A* [50,51,82] and *TNFRSF14* [29,50,51]. The pro-inflammatory interleukin 1 alpha (*IL-1A*) initiates immune response processes in the brain when it is released from the microglia [83]. IL-1A then (like *IL-1b*) activate the IL-1R1 signaling which, in turn, may trigger the downstream nuclear factor-kappa B (NFκB) and mitogen-activated protein (MAP) kinase pathways, both involved in innate and adaptative immune responses [84,85]. TNF receptor superfamily member 14 (*TNFRSF14*), a family member of the TNFRSF, is released by glial cells in the brain to trigger the production of proinflammatory cytokines and mediate neuroinflammation and cell death [86].

We extended our convergence analysis to evaluate if the immune-related genes reported in brain samples of animal models of PTSD were also identified in bulk-transcriptomic analyses in post-mortem brains of individuals with PTSD from the VA National PTSD Brain Bank [2,25]. We identified *BLNK*, *C1RL*, *CCL5*, *CFH*, *DOCK2*, *FOXO3*, *INPP5D*, *ITGAX*, *NFIL3*, *TNFRSF13C*, *FKBP5*, *CDK2*, *EMP1*, *ADAM12*, *TIMELESS* and *KIF5A* as convergent immune-related DEGs across species [2,25,49,50,51]. Chemokine (C-C motif) ligand 5 (*CCL5*) participates in the peripheral immune response, stimulates migration of phagocytes across the BBB (Blood Brain Barrier) during neuroinflammation, and can act as a pro-inflammatory chemokine in the brain impacting metabolic and neurotrophic processes [87,88]. Dedicator of cytokinesis 2 (*DOCK2*) is a microglia marker that regulates neuroinflammatory processes related to neurodegenerative disorders, such as Alzheimer’s disease [89,90]. *FOXO3* (Transcription factor forkhead box O-3) is a transcription factor that regulates the expression of IL-10 anti-inflammatory cytokine in innate immune cells like macrophages and dendritic cells [91,92]. Another convergent DEG across species is the stress-related *FKBP5* gene. *FKBP5* participates in the glucocorticoid (cortisol) inflammatory response to stress by mediating the GR translocation to the nucleus [93,94].

An interplay between inflammation, glial cells, and glutamate has been proposed in psychiatric disorders, suggesting that a dysregulated immune response in the brain may stimulate glial cells to release glutamate. The increased concentration of glutamate in the extra synaptic space may promote the aberrant activation of the ionotropic and glutamate receptor, which could result in synaptic dysfunction and loss [17,95,96]. In this review, microglia markers such as *DOCK2*, as well as neuronal ionotropic and glutamate markers such as *GRIN2B* (Glutamate Ionotropic Receptor NMDA Type Subunit 2B) and *SCN8A* (sodium voltage-gated channel alpha subunit 1) were observed as dysregulated in PTSD. These findings suggest that epigenetic regulation in PTSD may also impact the interplay between inflammation, glial cells, and glutamate and induce synaptic loss in the brain.

Epigenomic findings

In animal models of PTSD, a study using the social-stress model evaluated both transcriptomic and methylomic changes in brain, blood, and spleen tissues; it also showed the differential methylation of promoter regions of genes related to inflammation in the hemibrain (gene-level data not provided) [50,51]. Malan-Muller evaluated long noncoding RNAs (lncRNAs) in the hypothalamus and reported that mRNAs translated to 13 fear extinction-related proteins, which were predicted targets of the associated lncRNAs to PTSD [49]. In humans, no PTSD epigenomic studies have been conducted in brain tissue. A targeted epigenetic age study performed in 117 postmortem motor cortex samples from the VA National PTSD Brain Bank reported an interaction between rs9315202 SNP and the decreased expression of klotho (KL) longevity gene [97]. The gene product of KL is known to control the brain-immune system interface in the choroid plexus and to regulate autophagy in Alzheimer’s disease [98,99]. Evidently, more research is needed to identify brain-specific epigenomic alterations in PTSD.

Our research group has recently conducted a parallel profiling of neuronal-specific DNA methylation (5mC) and hydroxymethylation (5hmC) in the human postmortem orbitofrontal cortex (OFC) region from individuals with PTSD and controls collected at the VA National PTSD Brain Bank. We identified PTSD-associated significant CpG sites in immune-related genes: *IL-7R*, *CD34*, *CD8A* and *C1QL1* genes with differential 5mC; and *IL-4R*, *IL-15*, *IL-21-AS1*, *TLR5* and *TNFSF14* with differential 5hmC [Unpublished data]. The *CD8A* and *IL-15* genes were differentially expressed immune-related genes across central and peripheral tissues reported in animal models studies [50,51]. Our human neuronal-specific epigenomic results replicate findings observed in a rodent model of PTSD with cross-tissue overlap and suggest that epigenetic mechanisms may regulate gene expression patterns of immune-related genes.

### 3.3. Inflammatory Proteins Reported in PTSD

Both anti- and pro-inflammatory proteins have been implicated in PTSD [100,101,102,103]. Recent reviews (each including more than 50 studies) have supported an increase of pro-inflammatory cytokines and the reduction of anti-inflammatory signals in individuals with PTSD [7,104]. For instance, IL-1b, IL-6, and TNF-α are common cytokines increased in blood samples of individuals with PTSD [81,105,106,107,108,109,110]. Lower levels of plasma IL-4 levels in individuals with PTSD were consistent with changes observed at the epigenetic level [65]. Furthermore, a variation of the immune cell proportion (based on the ratio of CD4 helper/inducer cells and CD8 cytotoxic/suppressor cells) may also be implicated in the inflammatory dysregulation of PTSD. For instance, increased atypical NK (Natural Killer), reduction of regulatory T cells [111,112,113,114,115,116,117] and increased monocytes [118] cells have been reported in peripheral samples of PTSD. A study examining astroglia and neurotrophic markers in the plasma of 20 PTSD veterans and 20 age-matched healthy control veterans reported low levels of neurotrophic factors (BDNF and NGF-β) as well as increased levels of glial fibrillary acidic protein (GFAP), TNF-α and IL-6 in PTSD veterans. Plasma levels of matrix metalloproteinases MMP2 and MMP9, which play a role in the maintenance of the BBB integrity, are elevated in individuals with PTSD [119]. In the cerebrospinal fluid, a dysregulation of pro-inflammatory and anti-inflammatory cytokines was indicated in individuals with PTSD [120]. After the intramuscular administration of capsaicin, a compound with anti-inflammatory properties [121], an increase of IL-1B and delayed increase of IL-10 was observed in individuals with PTSD [120].

In the brain, a recent neuroimaging study using positron emission tomography (PET) evaluated the microglia marker translocator protein (TSPO) tracked by the [11C]PBR28 probe in the prefrontal-limbic region of 23 individuals with PTSD and 26 healthy controls. TSPO availability in the prefrontal-limbic region was lower in PTSD cases and negatively associated with PTSD symptom severity. Further, PTSD cases showed higher CRP levels in blood samples. These results suggest a deficient neuroprotective function and peripheral immune activation in PTSD [29]. This study evaluating central and peripheral samples confirmed the association of system-wide CRP levels with PTSD, previously reported in other studies [29].

Taken together, studies evaluating the inflammatory dysregulation of PTSD at the protein level show an impaired homeostasis of the central and peripheral immunological response in the pathophysiology of PTSD. Moreover, it supports the notion that these alterations at the protein level maybe mediated by changes in gene expression resulting from epigenetic modifications of immune-related genes.

### 3.4. PTSD, Comorbidities, and the Immune Response

PTSD is often comorbid with other mental and physical health disorders [122,123], possibly due to shared mechanisms of immune dysregulation. A PTSD transcriptomic study evaluating the effects of Body Mass Index (BMI) and sex [2,124] found that *IL-1B* is differentially expressed in PTSD males with high BMI, implicating *IL-1B* in the comorbidity of obesity and metabolic syndromes in PTSD [2,124,125]. Peripheral inflammation, mainly driven by IL-1β, TNF-α and IL-6, have been related to the disturbance of metabolic processes in individuals with schizophrenia (including insulin resistance, hepatic inflammation, and obesity) [126]. TNF-α and IL-6 have also been connected to PTSD [105,106,107,108,109,110], which suggest that these cytokines may simultaneously participate in PTSD comorbidities. Furthermore, PTSD GWAS studies, which have identified genetic variants mapping to immune-related genes, have also shown a genetic correlation between PTSD and asthma (rg = 0.49, P = 0.0002), a respiratory disease linked to the immune system and inflammation [3]. Cognitive impairment and dementia have also been related to PTSD [127,128,129,130,131,132]. In a proteomic analysis, *NCAN*, *BCAN*, *CTSS*, *MSR1*, *MDGA1*, and *CPA2* were reported as upregulated in individuals with PTSD and comorbid mild cognitive impairment [133]. Of these proteins, MSR1 and CTSS are also upregulated in a blood transcriptomic study of individuals with comorbid PTSD and Alzheimer’s disease [60,134,135,136,137]. These findings suggest that dysregulation of the immune system may be a shared mechanism of PTSD and concomitant mental and physical disorder comorbidities, and this highlights novel therapeutical interventions targeting the immune signaling for the treatment of PTSD, as well as associated comorbidities [126,138,139].

### 3.5. Deciphering Systemic Immune Response in PTSD Mediated by Epigenetic and Transcriptomic Changes

Integrative multi-omics analyses are an excellent strategy to identify and prioritize molecular markers that can help inform disease risk, prognosis, and treatment interventions [140,141,142]. Further, a comprehensive evaluation of multi-omics domains in a dimensional manner reveals convergent biological pathways across multiple layers and assesses their interactions at the molecular and circuit levels [140,143,144]. In this review, we show that cross-species studies at the genomic, epigenomic, transcriptomic, and proteomic level support a system-wide dysregulation of the immune signaling in PTSD [7,138,145]. Here, we conducted an integrative multi-omics analysis of the literature in PTSD across the central and peripheral systems in both human and animal models, which revealed 84 convergent immune-related genes (Appendix A).

Annexin A2 (*ANXA2*) was upregulated in the dlPFC of human postmortem brains [2] and a DEG in animal models of PTSD including the hypothalamus [49], hippocampus, amygdala, mPFC, hemibrain, blood, and spleen [50,51]. *ANXA2* is expressed on the surface of several innate immune cells such as dendritic cells, macrophages, and monocytes; and its anti-inflammatory and pro-inflammatory activities vary among acute and chronic stages of inflammation [146]. *ANXA2* facilitates wound healing orchestrating matrix remodeling, membrane repair, angiogenesis, vesicle fusion, and cytoskeletal organization [147]. Adenosine A2a receptor (*ADORA2A*) is also a dysregulated gene in the hypothalamus [51] and the hippocampus, amygdala, mPFC, hemibrain, blood and spleen [50,51] from animal models of PTSD. *ADORA2A* was also identified in our neuronal-specific epigenomic analysis in post-mortem orbitofrontal samples of individuals with PTSD [Unpublished data]. The gene product of *ADORA2A* is a receptor involved in the modulation of the response of microglia and astrocytes during neuro-inflammation [148].

The TNF signaling was a convergent dysregulated pathway across all approaches (central and peripheral tissues, with epigenomic and transcriptomic dysregulation and human and animal models of PTSD). At the protein level, TNF-α was increased in plasma samples of individuals with PTSD [119]. At the transcriptomic level, *TNFRSF14* and *LTB* (also known as tumor necrosis factor C) were convergent downregulated genes across transcriptomic analyses performed in peripheral tissues of both species. Furthermore, *TNFRSF14* and *TSPOAP1* had a decreased gene expression in females with PTSD [2]. At the epigenomic level, in our ongoing neuronal-specific study, we identified a hyper hydroxymethylated CHH site in the intergenic region between *CD70* and *TNFSF14* genes (chr19:6614775), which has a predicted transcription factor binding site to ZNF135. These multi-omic findings suggest an important role of the TNF pathway in PTSD.

Based on the protein-protein interaction analysis of convergent genes in the central and peripheral tissues using STRING [74], significant enriched pathways (FDR < 0.05) include cytokine-cytokine receptor interaction (*OSM*, *IL-2*, *IL-1RN*, *TNFRSF13C*, *CXCL3*, *CCR1*, *CXCL10*, *TGFBR1*, *IL-6*, *IL-10*, *CXCL2*, *IL-12RB1*, and *CCL5*), NF-kappa B signaling pathway (*BLNK*, *PRKCQ*, *ZAP70*, *TNFRSF13C*, *CXCL3*, *BCL2L1*, *TLR4*, *MYD88*, and *CXCL2*), Toll-like receptor signaling pathway (*CXCL10*, *TLR1*, *TLR4*, *IL-6*, *TLR6*, *MYD88*, *PIK3R1*, and *CCL5*), JAK-STAT signaling pathway (*OSM*, *IL-2*, *BCL2L1*, *GRB2*, *IL-6*, *IL-10*, *PIK3R1*, and *IL-12RB1*), Natural killer cell mediated cytotoxicity (*ZAP70*, *GRB2*, *NFATC2*, and *PIK3R1*), and Th17 cell differentiation (*IL-2*, *PRKCQ*, *ZAP70*, *TGFBR1*, *NFATC2*, *IL-6*, and *IL-12RB1*). 

Brain-specific protein markers were dysregulated in the plasma of individuals with PTSD [119]. Brain cell-type specific gene expression patterns of human and murine RNA expression data sets were recently reported showing the top 1000 cell type-specific genes from astrocytes, oligodendrocytes, microglia, neuron and endothelial cells [149]. Taking advantage of a recent study reporting brain cell type markers and based on 84 convergent genes recovered in this review, we explored if brain cell type specific genes may be affected by epigenomic and transcriptomic changes in PTSD across central and peripheral systems in human and animal models. We found that ~50% of convergent genes were enriched for cell type markers in the brain. *CXCL3* is enriched in astrocyte cells; *PRKCQ*, *SEMA4D*, *LPAR1* and *SHC4* in oligodendrocytes cells. Also, *C1QA*, *C1QB*, *CCR1*, *IL-10*, *IL-1RN*, *NFATC2*, *OSM*, *PLXDC2*, *TGFBR1*, *TLR1*, *TLR4*, *AKAP13*, *MILR1*, *TLR6*, *TMEM106A*, *BLNK*, *DOCK2*, *TNFRSF13C*, *ITGAX*, *INPP5D*, *CTSS*, *GRB2* and *CHSY1* were enriched for microglial cells, *CXCL2*, *IL-6*, *LTF*, *IFI6*, *EPSTI1*, *CDK2*, *ANXA2*, *CFH*, *EMP1*, *COL1A1*, *CYP1B1*, *COL4A1*, *IGFBP4*, *COL6A2* and *RCSD1* for endothelial cells, and *WIPF3* for neuronal cells [149]. By using flow-sorted PTSD transcriptomic data from the blood [68], we evaluated whether these convergent genes enriched for brain cell type markers are also differentially expressed in peripheral cells such as monocytes, CD4, and CD8 cells. *PXK*, *TAOK1* and *SET* (enriched for oligodendrocyte cells), and *CNOT1* and *NCKAP1L* (enriched for microglial cells) were PTSD DEGs reported in monocytes. Furthermore, *FGD4*, *PLXDC2*, *AKAP13* and *CD163* (enriched for microglial cells) were DEGs in CD4/CD8 cells [68].

The multiomic convergence of the immune-related genes summarized in this review support a system-wide dysregulation of the immune signaling in PTSD and reinforce the immunological systemic dysregulation as a hallmark of PTSD [7,138,145]. Innate immunity plays an important role in the adaptative response to physiological or psychological factors provoking the release of both anti and pro-inflammatory molecules which are mediated by macrophage, mast cells, and polymorphonuclear leucocytes in peripheral tissues, as well as microglia in the brain [150,151,152]. Innate immune response upon cellular stress may be activated by damage-associated molecular patterns (DAMPs), endogenous molecules that are released by injured tissues. Toll-like receptors (TLRs, one of the cross-tissue convergent pathways identified here) recognize DAMPs which, in turn, activate the NF-κB pathway (another convergent pathway across tissues identified here) and induce cytokines releasing [151,153,154]. A homeostatic immune signaling in both peripheral and central tissues is necessary to maintain its protective function and reduce the risk of developing PTSD [152,155]. In contrast, a maladaptive immune response, possibly mediated by glucocorticoids [9,11,12,13,14,15], may have the opposite effect, reflected by the activation of microglia, neuronal death, accelerated neurodegenerative processes, as well as a disruption of the BBB structure allowing peripheral immune cells into the brain [152,156,157].

The intercommunicative nature of immune signaling across central and peripheral tissues is observed in healthy brains where IL-6 and IL-1 (released by glial cells) are able to activate the immune signaling in both brain and periphery [150]. In PTSD, a study evaluating microglial markers in the brain and CRP in peripheral tissues suggested that inflammatory suppression in the brain is related to increased peripheral inflammatory response [29]. In contrast, an animal model study evaluating acute and chronic stress responses in both central and peripheral tissues suggests a sustained systemic inflammatory response in PTSD resulting in tissue damage in peripheral tissues and the inhibition of synaptic plasticity and neurogenesis in the brain [50,51].

Some of the identified convergent molecules in this review support the hypothesis of an increased peripheral inflammation in PTSD. The transcriptomic dysregulation of immune cells (i.e., monocytes, CD4T, CD8T and B lymphocytes) in PTSD involves the upregulation of *FKBP5* and *PI4KAP1* across all peripheral cell types [68]. IL-6, with both anti and pro-inflammatory functions, has been a consistent finding at multiple levels in the increased peripheral inflammation of PTSD [7,16]. At the transcript level, IL-6 was reported as upregulated in two transcriptomic studies, one conducted in blood samples of PTSD individuals [62] and one in animal models examining both central and peripheral tissues [50,51]. At the epigenetic level, no studies have reported differential methylation of IL-6 in PTSD. At the protein level, IL-6 was increased in blood samples of PTSD [119]. Elevated serum levels of IL-6 in women with PTSD was related to dysregulation of gene expression of other inflammatory molecules such as IL-4 and IL-18R1 [54]. Additional PTSD peripheral studies of pro-inflammatory molecules have shown increased levels of TNF-α in PTSD veterans [119], an upregulation of IL-1, IL-1B, IL-8, IL-4R, IL-16 and NF-κB in military personnel [62], increased levels of CRP associated with PTSD symptom severity [29], as well as an association between the *CRP* polymorphism (rs3091244)and lifetime trauma exposure and PTSD severity and the association between rs1205 and rs2794520 with PTSD severity and serum CRP levels relation [6,30].

In brain tissue, animal model studies show an increase of IL-6 related to PTSD [16]. The upregulation of IL-6 across central and peripheral tissues has only been reported in an animal model study [50,51]. In the human brain, IL-6 is known to be involved in neurogenesis and oligodendrogenesis during early postnatal developmental stages [158,159]. However, in PTSD, no studies examined in this review show IL-6 dysregulation in the brain. A recent human study supports brain inflammatory suppression in PTSD based on low brain availability of TSPO, and decreased expression of *TNFRSF14* and *TSPOAP1* in the postmortem brains of a female subgroup with PTSD [29]. A downregulation of these *TNFRSF14* and *TSPOAP1* genes has also been found in an animal model study [50,51]. In addition, reduced expression of the proinflammatory IL-1A was observed in the PFC of individuals with PTSD [82] and in an animal model [50,51]. *FKBP5* and *NR3C1*, both anti-inflammatory molecules [156], were downregulated in the medial mPFC and amygdala of animal models, respectively [50,51]. In contrast, a study in postmortem dlPFC samples of individuals with PTSD found an upregulation of *FKBP5* [2]. Similarly, upregulation of *FKBP5* in the amygdala was reported in another animal model study of PTSD-like behavior [160].

Overall, the studies reviewed here reinforce the hypothesis of an increased peripheral inflammation and a suppressed neuroinflammation in PTSD. However, given the somewhat inconsistent findings in the literature, more research is needed to better characterize the immune dysregulation of PTSD in the brain. Several limitations of the current literature include the lack of evaluation of varying trauma types as well as the duration of trauma exposure. It is important to also note the high heterogeneity of the PTSD’s phenotypic definition used across the human studies, from using case-control binary traits to symptom-level continuous traits. In the animal work, there is an inherent limitation in mimicking socio-cultural environmental exposures observed in individuals with PTSD [24] and recapitulating the complex clinical outcomes of PTSD and related comorbidities. Lastly, studies are needed to directly evaluate the role and possible interplay of anti- and pro-inflammatory mechanisms in the PTSD immune dysregulation.

In conclusion, the convergent cross-tissue immune-related genes and pathways identified here in both human and animal models support a systemic immune dysregulation in PTSD and a potential crosstalk between the central and peripheral systems (Figure 2).

### 3.6. Drug Development

Substantial evidence supports an important role of a systemic immune dysregulation in PTSD, opening new avenues of drug development for treatment interventions. A previous study showed that immune-related genes, specifically the pro-inflammatory macrophage imbalance (M1/M2 ratio), can serve as a biomarker of response to psychotherapy in the treatment of individuals with PTSD [161]. Therapeutical properties of anti-inflammatory drugs have been also evaluated in animal models of PTSD. For instance, minocycline, an antibiotic with anti-inflammatory properties, reduced the level of IL-1, IL-6 and TNF-α and ameliorated anxiety behaviors after trauma exposure [162]. Here, we performed a drug repurposing analysis using the Drug-Gene Interaction Database (DGIbd) [163] to identify druggable gene targets as potential therapeutic pharmacological treatments [164]. The DGIbd analyzes drug-gene interactions and assigns an interaction score for each pair of gene-drug pairs to help prioritize gene-level data. We conducted a drug repurposing analysis for the 84 identified convergent genes across tissues and species in PTSD (applying query score 5 [163]. We found that convergent genes enriched in Th17 differentiation in the protein-protein interaction analysis (mentioned before)could be a possible target mechanism for drug action. In this analysis, we identified drug targets for genes involved in three main pathways: the toll-like receptor signaling pathway (CXCL10, IL-6, and TLR4), Th17 cell differentiation (ZAP10, IL-6, and TGFBR1), and cytokine-cytokine receptor interaction (CCR1, CXCL10, IL-6, TGFBR1) (Appendix A). Drug discoveries are based on their target on monoclonal antibodies for the inhibition of immune proteins, including siltuximab, olokizumab, and clazakizumab (IL-6), vactosertib (TGFBR1), and petesicatib, and odanacatib (Cathepsin S). More research is needed to evaluate the effects of these drugs in the treatment of PTSD.

### 3.7. Limitations, Future Directions, and Concluding Remarks

A growing body of research in humans and animal models supports the dysregulation of the immune system as an important mechanism in PTSD. Here, we reviewed the recent PTSD literature of the genomic, epigenomic, transcriptomic, and proteomic studies to identify the dysregulated immune factors in the central and peripheral systems of animal models and humans implicated in PTSD. We reveal a system-wide immune dysregulation on PTSD and characterize a potential crosstalk between the central and peripheral immune systems. However, the immunological dysregulation in PTSD can be highly heterogeneous, acting on both pro-inflammatory and anti-inflammatory signaling pathways. This heterogeneity may be driven by differences in the duration, type, and intensity of the trauma, but also by potential cell-type specificity with different pathophysiological functions in the acute and long-lasting effects of trauma response which is essential for the development and maintenance of PTSD. Of note, PTSD is a complex multifactorial disorder with high phenotypical heterogeneity. Research aimed to disentangle the role of the immune system in the different domains or symptoms of PTSD is greatly needed. Furthermore, we also highlight the need to study the role of the immune system in the brain in the context of PTSD, particularly in humans. This review sheds new light into the convergent roles of the central and peripheral immune dysregulation in PTSD and identifies potential novel drug targets for treatment development.

## Figures and Tables

**Figure 1 biomedicines-10-01107-f001:**
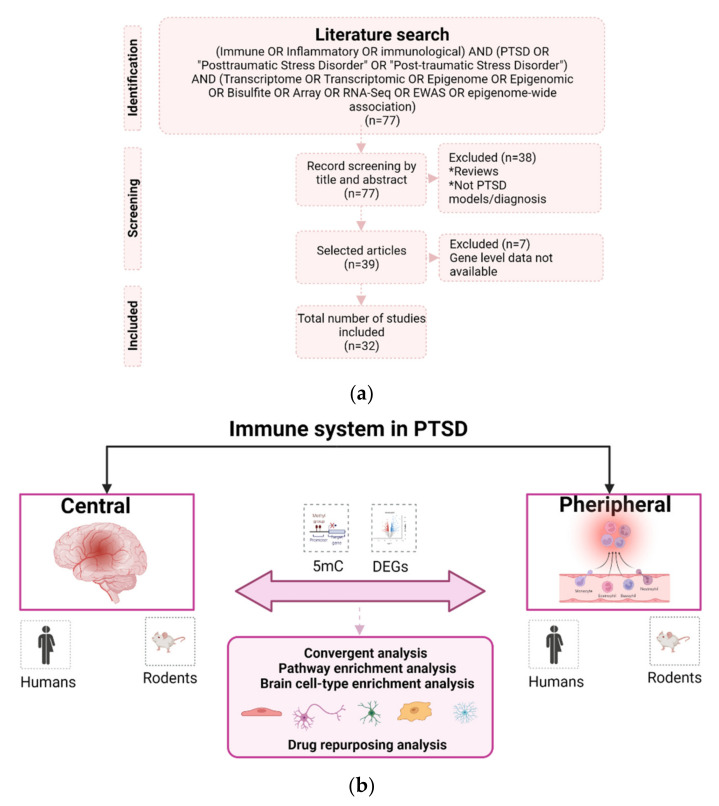
(**a**) Search and selection procedure of the literature on the immune system in PTSD. By screening by title and abstract, we selected a total of five papers conducted on brain tissues and 27 in peripheral tissues across species. These studies assessed immune-related markers at distinct levels: genetic variants conferring risk to PTSD, transcriptomic and epigenomic changes that occurred after traumatic exposure or associated with trauma exposure or PTSD, and the dysregulation of inflammatory proteins related to the pathophysiology of PTSD. (**b**) After the selection of immune-related genes reported in the studies, we explored the convergence across species in both central and peripheral tissues in terms of regulatory level (5mC: methylation and DEG: Differentially Expressed Genes). We then explored the convergence between central and peripheral tissues and conducted a GO enrichment analysis, brain cell-type enrichment analysis and drug repurposing analysis to identify potential markers involved in the systemic immune dysregulation in PTSD as well as potential treatments.

**Figure 2 biomedicines-10-01107-f002:**
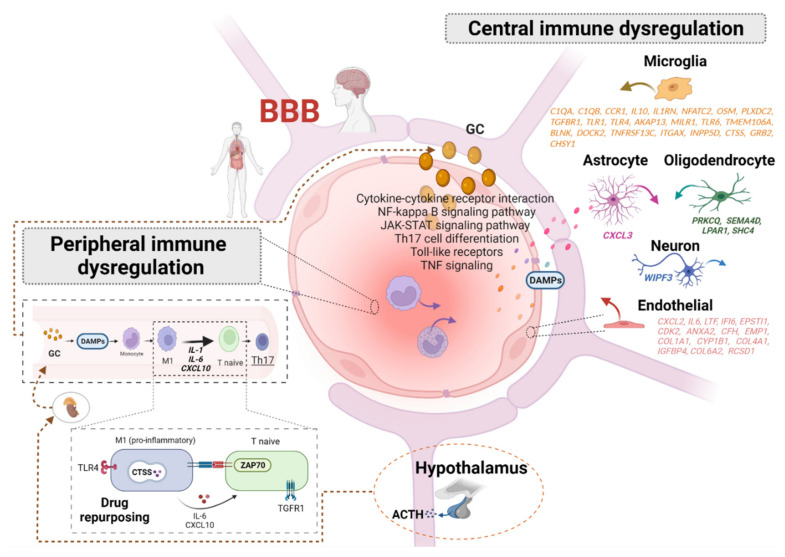
Depicting systematic immune response in PTSD. The immune signaling pathway is dysregulated in PTSD at multiple regulatory levels. Recent whole genome analysis in PTSD conducted on both human and animal models of PTSD have demonstrated that epigenomic and transcriptomic changes may mediate the development, maintenance, and clinical presentation of PTSD. HPA axis releasing ACHT stimulates cortisol (human (GC) glucocorticoids) production which, in turn, also mediates gene expression and exacerbates immune response in the periphery and brain. In this review, 84 genes that participate in immune signaling processes were commonly reported as differentially expressed or methylated genes in central (brain) and peripheral (such as blood and saliva) tissues across species. ~50% of those convergent genes correspond to brain-cell type specific markers such as astrocytes and microglia, which are discriminated by color. Upon inflammatory response, the blood brain barrier (BBB) may be affected and allow a higher migration of molecules crossing the brain and periphery. Thus, based on the literature review, we propose the existence of common regulatory markers able to cross BBB, impact the expression of those 84 convergent genes and orchestrate systemic immune dysregulation in PTSD through important immunological pathways such as cytokine-cytokine interaction, TNF and NF-κB/JAK/STAT signaling. These common regulatory markers may correspond to damage-associated molecular patterns (DAMPs) which are released from damaged cells and activate the innate immune response. By using drug repurposing analysis, we also suggest that Th17 differentiation may be a possible target mechanism for drug action, as well as Toll-like receptor signaling and cytokine-cytokine receptor interaction, for the treatment of PTSD.

## Data Availability

Not applicable.

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
