# Peer review of "Central and Peripheral Immune Dysregulation in Posttraumatic Stress Disorder: Convergent Multi-Omics Evidence"

_biomedicines, 2022, doi:10.3390/biomedicines10051107_

Round 1
Reviewer 1 Report
The paper reviews the role of the inflammatory system in PTSD across tissue and species by using an integrative multi-omics approach with a particular focus on the genomics, transcriptomics, epigenomics, and proteomics domains and also proposes potential novel drug targets for PTSD treatment.
The review gives a comprehensive overview of potential interplay between the central and peripheral immune system and their convergent roles in PTSD. The manuscript is well written, the analysis of the data is well performed, the list of references is adequate. In my opinion the paper represents a significant contribution to the field and therefore I commend it for publication.
Author Response
We thank the reviewer for these comments
No concerns were noted

Reviewer 2 Report
The review by Núñez-Rios is focused on an interesting topic. However, there is no clear story and in the present form, the manuscript is more of a list of immune system molecules changed in PTSD than showing the relevant connection between molecules involved. Moreover, there is no discussion on the results of studies presented in the manuscript which again emphasizes the descriptive nature of this review.
I have the following comments:
Major:
1) Please add a Discussion on the presented results emphasizing the differences between CNS and periphery.
2) Please add the list of abbreviations. In the present form, it is not convenient to search for the first appearance of the abbreviation.
3) Redesign Figure 2: cells shown on the left (down) should be identical to the detail above.
Minor:
1) If the journal has the policy to cite the references as [number], it is not appropriate to cite (line 130)...Sheerin et al. studied rare....Please change.
2) line 395: TNF-a should be read TNF-alpha(in greek)
Author Response
1) Please add a Discussion on the presented results emphasizing the differences between CNS and periphery.
We thank the reviewer for this helpful suggestion and agree that adding a discussion on the differences between CNS and periphery will add value, resulting in an improved manuscript.
In the section 3.5 of the revised manuscript titled “Deciphering systemic immune response in PTSD mediated by epigenetic and transcriptomic changes” we have added a discussion on the role of the central and peripheral immune signaling in trauma response and PTSD and the difference between the two. We also pinpoint the inconsistencies in the literature and highlight the phenotypical heterogeneity of PTSD and high methodological variability in animal models as potential driving factors of such inconsistencies.
2) Please add the list of abbreviations. In the present form, it is not convenient to search for the first appearance of the abbreviation.
As suggested by the reviewer, we have added a list of abbreviations after the ‘Limitations, future directions, and concluding remarks’ section in the revised manuscript.
3) Redesign Figure 2: cells shown on the left (down) should be identical to the detail above.
We have redesigned Figure 2 following reviewer’s recommendations.
Minor:
1) If the journal has the policy to cite the references as [number], it is not appropriate to cite (line 130)...Sheerin et al. studied rare....Please change.
We thank the reviewer for this correction. We have edited the manuscript by citing all references as [number].
2) line 395: TNF-a should be read TNF-alpha(in greek)
We have edited the manuscript by changing “TNF-a” to “TNF-a”

Reviewer 3 Report
Núñez-Rios et al., biomedicines 2022
The authors conducted a systematic review of recent studies employing transcriptomic, epigenomic, or proteomic approaches to assess PTSD-associated changes in humans or animal models. As immune system dysfunction has been implicated in stress disorders like PTSD, the authors summarized the immune-related changes described in these studies and identified the changes that were reported in more than one studies. The authors discussed the immune-related changes at the gene, transcript, epigenome, and protein levels in both the periphery and the brain. Finally, the authors identified several specific immune pathways that are likely associated with PTSD and suggested directions for future therapeutic development. Convergent genes that are identified across multiple studies and species are likely of high importance which could facilitate future mechanistic studies and drug development.
Major point:
- Throughout the manuscript, the authors implied immune system dysfunction as a contributing factor to the etiology of PTSD. However, most studies conducted binary comparisons between humans with and without PTSD. Differentially regulated genes/markers identified in this manner may not necessarily indicate a role in the development of the disorder. The authors should re-evaluate the primary studies and revise their statements accordingly.
Minor points:
- One of the inclusion criteria is “studies using wild-type and drug-free animal models of PTSD”. The authors should elaborate the reason for this criterion. Are wild-type and drug-free animal models of PTSD more similar to human PTSD?
- Throughout the result section, the authors summarized a large number of gene symbols, which are challenging for the readers to keep track of. To make the manuscript more readable to the general audience, it would be helpful to use tables to include full gene names and brief descriptions of gene function or pathways involved. This would be particularly helpful for the readers to interpret the convergent genes.
- Line 312: The authors should define epigenetic age.
- Line 536-537: The authors should explain how the drug repurposing analysis was conducted.
Author Response
Major point:
1) Throughout the manuscript, the authors implied immune system dysfunction as a contributing factor to the etiology of PTSD. However, most studies conducted binary comparisons between humans with and without PTSD. Differentially regulated genes/markers identified in this manner may not necessarily indicate a role in the development of the disorder. The authors should re-evaluate the primary studies and revise their statements accordingly.
We thank the reviewer for this helpful suggestion. Since our review was designed to identify immune-related genes in PTSD that may participate in the wide phenotypical heterogeneity of this disorder, we observed that selected papers utilized varying study designs in the phenotypic definition of PTSD (e.g.,
case-control vs. continuous symptom-level variables as well as different trauma types and duration of trauma exposure). As a result, and acknowledging reviewer’s suggestion, we have revised our manuscript by removing the term “etiology”. We also added this as a limitation in the section titled “Deciphering
systemic immune response in PTSD mediated by epigenetic and transcriptomic changes” in the revised manuscript.
1) One of the inclusion criteria is “studies using wild-type and drug-free animal models of PTSD”. The authors should elaborate the reason for this criterion. Are wild-type and drug-free animal models of PTSD more similar to human PTSD?
The aim of this study is to review the immune dysregulation in PTSD by evaluating genomic, transcriptomic, epigenomic, and proteomic studies. By considering “studies using wild-type and drug-free animal models of PTSD” as an inclusion criterion, it helps us identify biological changes that are specific
to the trait of interest rather than those resulting from pharmacological agents used in the studies. We have added this rationale to the revised manuscript.
2) Throughout the result section, the authors summarized a large number of gene symbols, which are challenging for the readers to keep track of. To make the manuscript more readable to the general audience, it would be helpful to use tables to include full gene names and brief descriptions of gene function or
pathways involved. This would be particularly helpful for the readers to interpret the convergent genes.
In the revised manuscript, we have added a table including the full gene names and a brief description of gene function or pathways involved. This table is included as Supplementary Table 2.
3) Line 312: The authors should define epigenetic age.
We have added a definition of epigenetic age in Line 321-323 of the revised manuscript.
4) Line 536-537: The authors should explain how the drug repurposing analysis was conducted.
We added a description of the drug repurposing analysis in Lines 626-630 of the revised manuscript

Reviewer 4 Report
The manuscript from Núñez-Rios DL et al. provided a comprehensive review of the relationship between immune dysregulation and PTSD in a convergent multi-omics approach. Generally, the manuscript was well-written and covered all necessary aspects. There are only a few minor typos that require correction before publication, as noted below:
Line 94-98: there is a mixed use of straight quotes and curled quotes, please make consistent;
Line 124 and all following cases: in "n=429", n should be italic;
Line 173 and all following cases: there is a mixed use of NFκB and NFkB, please make consistent;
Section "Epigenomic", starting line 228: human gene names should be italic;
Line 385: Smith et al., missing a full stop;
Author Response
Few minor typos:
1) Line 94-98: there is a mixed use of straight quotes and curled quotes, please make consistent;
We have revised the manuscript accordingly.
2) Line 124 and all following cases: in "n=429", n should be italic;
We have revised the manuscript accordingly.
3) Line 173 and all following cases: there is a mixed use of NFκB and NFkB, please make consistent;
We have revised the manuscript accordingly.
4) Section "Epigenomic", starting line 228: human gene names should be italic;
We have revised the manuscript accordingly.
5) Line 385: Smith et al., missing a full stop;
We have edited all references following the [number] format throughout the manuscript

Round 2
Reviewer 2 Report
All points weew solved.